# Specific Emitter Identification Method for Limited Samples via Time–Wavelet Spectrum Consistency

**DOI:** 10.3390/s25030648

**Published:** 2025-01-22

**Authors:** Chunyang Tang, Jing Lian, Li Zheng, Rui Gao

**Affiliations:** 1School of Electronic and Information Engineering, Lanzhou Jiaotong University, Lanzhou 730070, China; lian322scc@163.com (J.L.); zhengli@lzjtu.edu.cn (L.Z.); gaorui@lzjtu.edu.cn (R.G.); 2Silk Road Fantian (Gansu) Communication Technology Co., Ltd., Lanzhou 730070, China

**Keywords:** specific emitter identification, comparative learning, data augmentation, deep learning, continuous wavelet transform

## Abstract

Specific emitter identification (SEI) is a technique that identifies the emitter through physical layer features contained in radio signals, and it is widely used in tasks such as identifying illegal transmitters and authentication. Thanks to the development of deep learning, SEI tasks based on deep learning have achieved significant improvements in recognition performance. However, illegal transmitters often broadcast for short durations and at low frequencies, resulting in very limited available training samples. In such cases, directly training deep learning models may lead to underfitting issues, thereby reducing recognition accuracy. In this paper, to address the issue of traditional methods struggling to classify transmitters when there is a scarcity of emitter samples and limited training data, we propose a method based on TFC-CNN. We propose a TFC-CNN method. Specifically, we first use continuous wavelet transform (CWT) as a data augmentation method to construct time–wavelet spectrum sample pairs. Then, we use complex-valued neural networks (CVNNs) and deep convolutional neural networks (DCNNs) to extract the hidden emitter identity features from the time and wavelet spectrum samples. We train the model using the normalized temperature-scaled cross-entropy (NT-Xent) loss and cross-entropy (CE) loss, ensuring the consistency of feature vector distributions across the two modalities with cosine loss. Finally, we fine-tune the model to achieve SEI tasks with few samples. Experimental results on open-source WiFi datasets and automatic dependent surveillance–broadcast (ADS-B) datasets show that our proposed method outperforms existing state-of-the-art methods. With only 5% of the training samples, the recognition accuracy for the ADS-B test dataset is 84.10%, and for the WiFi test dataset, it is 96.99%.

## 1. Introduction

With the development of wireless communication technology, wireless devices have been widely applied in various fields such as intelligent transportation, smart factories, and the Internet of Things (IoT) [1,2]. Billions of wireless communication devices broadcast and receive information in free space, bringing great convenience to people’s daily lives while also raising concerns about data security. Current data security relies on cryptographic methods; if passwords are cracked or subjected to malicious attacks, data security will be threatened [3]. Therefore, developing a recognition method that does not depend on specific communication protocols and a more secure and reliable identity recognition technology becomes particularly important [4]. As a physical layer-based identity authentication technology, SEI shows great potential in this regard.

Specific emitter identification, also known as radio frequency fingerprinting, originates from the physical defects of emitter hardware devices [5], typically manifested as phase noise and in-phase and quadrature (IQ) offset caused by quadrature modulation [6], carrier frequency offset produced by up-converters [7], and nonlinear distortions generated by power amplifiers [8]. These features are stable and unique, capable of characterizing the identity of the transmitter. Since SEI is based on identifying hardware defects for transmitter identity authentication, it is difficult to tamper with or deceive at the software level, effectively ensuring data security [9].

Traditional specific emitter identification methods mainly construct emitter features through signal processing techniques and classify them using machine learning methods such as regression analysis [10], maximum likelihood classification [11], multilayer perceptron, support vector machines [12], and linear Bayesian classifiers. Although these machine learning classification methods can achieve specific emitter identity recognition, manually constructed features struggle to represent emitter characteristics, and machine learning classifiers suffer from weak generalization ability and low accuracy [13]. Consequently, with the development of deep learning technology, traditional specific emitter identification methods have gradually been replaced by SEI methods based on deep learning.

Deep learning-based SEI identification methods mainly consist of two parts: feature extraction and classification networks. For feature extraction, common methods include approximate entropy [14], short-time Fourier transform (STFT) [15,16], cosine spectra [17], variational mode decomposition [18], fractal dimensions [19], Wigner–Ville distribution [20], and power spectra [21]. These feature extraction methods reduce channel noise and interference from modulated data on specific emitter features, increasing the accuracy and stability of SEI. The above methods primarily use traditional signal processing algorithms to preprocess specific emitter identification, with some scholars optimizing traditional algorithms specifically for SEI tasks. L. Peng et al. [22] introduced a feature extraction method based on differential constellation trace figure (DCTF), which forms a 2D image similar to a constellation plot after differentiating IQ for network input, achieving good performance in SEI tasks. In recent years, various optimization methods for DCTF have been widely proposed [23,24,25,26]. Y. Tu et al. [27] proposed combining short-time Fourier transform, wavelet transform, time, and Wigner–Ville transform into four modalities for SEI in real IoT environments. Combining multiple modalities as emitter features improves SEI accuracy but also increases computational complexity. For classification networks, deep learning models such as generative adversarial networks [28], convolutional neural networks [29], complex Fourier neural network [30], autoencoders [31], and complex-valued neural networks [32] are used for training. These classification networks achieve good recognition performance in SEI tasks, with some scholars proposing to incorporate transformer and attention modules into the network to enhance performance. A. Al-Shawabka et al. [33] used a CNN network to perform SEI tasks on 300,000 transmissions from 10,000 WiFi and ADS-B devices, achieving a recognition accuracy of 77% in large-scale device scenarios. This verifies that SEI technology can still perform well when facing large-scale IoT devices.

However, for non-cooperative emitter identification, obtaining sufficient samples is very challenging. When deep learning-based SEI methods lack enough samples for training, the network model inevitably faces issues such as underfitting, leading to a significant decrease in SEI accuracy. To address limited sample sizes for deep learning training, existing methods include data augmentation [34], contrastive learning [35], generative adversarial networks [36], and meta-learning [37]. This paper mainly achieves small-sample emitter identification through a combination of data augmentation and contrastive learning. Two typical contrastive learning networks, SimCLR [38] and Siamese [39], primarily build sample pairs through image transformations and use contrastive loss to bring features of the same class closer together in feature space while pushing features of different classes apart. Y. Peng et al. [40] used rotation, flipping, adding Gaussian noise, and shifting as data augmentation methods and employed supervised contrastive learning to achieve SEI with a small number of samples, achieving an accuracy of 92.68% with only 5% of the samples. Y. Wang et al. [41] constructed a contrastive network model using CVNN, demonstrating the advantages of contrastive learning networks in generalization under small-sample SEI. Z. Wu et al. [42] used signal translation invariance as a data augmentation method and trained the model with contrastive loss, achieving over 90% recognition accuracy.

Contrastive learning can achieve SEI with a small number of samples; however, existing data augmentation methods mainly add random disturbances to signals without bringing additional effective information. This paper uses CWT as a data augmentation method, providing different transformed domain data to the contrastive learning network. However, time-domain signals and wavelet spectrum data belong to different modalities—vectors and images. Therefore, this paper draws inspiration from ActionCLIP [43] to use different networks for processing multimodal data and trains the network with contrastive loss.

Overall, the main contributions of this paper are as follows:We propose a data augmentation method based on TWC that leverages traditional CWT algorithms to construct multimodal information, which serves as an auxiliary dataset to expand the sample size of the training dataset. This method can still fully train deep learning models and achieve specific emitter identification tasks even with a small number of samples.We introduce an SEI framework based on contrastive learning, utilizing CVNN and DCNN to extract feature vectors from wavelet spectrograms and IQ vectors, respectively. The vector-mask module adjusts the probability distributions of feature vectors extracted from wavelet spectrograms and IQ vectors. Finally, we combine the two modalities through contrastive learning and supervised learning to improve SEI recognition accuracy.The proposed SEI method based on TWC-CNN is validated on ADS-B and WiFi datasets. The experimental results show that it outperforms the six most advanced methods in the small sample scenario.

The remainder of this paper is organized as follows. Section 2 describes the basic knowledge of the small-sample specific emitter identification problem that needs to be addressed in this paper. Section 3 describes the components of the time–wavelet spectrum consistency convolutional neural networks (TWC-CNNs) proposed in this paper, including the complex convolutional neural network, the TWC data augmentation method, and the loss function combining contrastive loss and cross-entropy loss. The evaluation experiments are conducted in Section 4, along with an analysis of the experimental results. The conclusion is drawn in Section 5.

## 2. Problem Description

### 2.1. Specific Emitter Identification Model

The formation principle of the radio frequency fingerprint in the emitter is shown in Figure 1. The modulation module and analog-to-digital conversion module in the transmitter convert the baseband signal into a modulated digital signal. Then, the digital upconversion module multiplies the modulated digital signal with the local oscillator signal and filters out high-frequency information through a low-pass filter. Afterward, the local oscillator module shifts the signal to the radio frequency band. Finally, the power amplifier module amplifies the signal, which is then radiated into free space by an antenna.

However, the digital upconversion module, local oscillator module, and power amplifier module in the transmitter hardware system are all physical devices that inevitably have defects in the manufacturing process. These defects can cause signal distortion, such as phase noise and IQ offset caused by the digital upconversion module [6], carrier frequency offset caused by the local oscillator module [7], and nonlinear distortion caused by the power amplifier module [8]. These hardware defects introduce regular distortions to the signal’s frequency, phase, and amplitude.

In the process of signal transmission, the local oscillator module and digital upconversion module of the transmitter shift the original signal x(t) to the radio frequency band, causing phase noise, IQ offset, and carrier frequency offset. Signal generation introduces distortions in frequency and phase y(t):(1)$$\begin{array}{cc} y(t) & =x(t)e^{2\pi j(f+f^{\prime }(t))t+\varphi_{0}+\varphi (t)}\\ & =x(t)e^{2\pi jft+\varphi_{0}}e^{2\pi jf^{\prime }(t)t}e^{\varphi (t)} \end{array}$$

In Formula (1), f represents the carrier frequency, $f^{\prime }$ represents the carrier frequency offset, φ(t) represents phase noise and IQ offset, and $\varphi_{0}$ is the initial phase. The distorted signal can be decomposed into the product of the original signal, carrier signal, phase difference $e^{\varphi (t)}$, and rotation angle $e^{jw^{\prime }(t)}$. Phase difference and rotation angle manifest as phase noise, IQ imbalance, and IQ rotation in the received baseband signal constellation diagram.

During signal transmission, for an ideal power amplifier module, the output voltage varies linearly with the input voltage. However, the actual input–output response curve of the amplifier shows significant nonlinearity in both saturation and non-saturation regions, leading to high-order harmonics in the output signal. For a single-tone sinusoidal signal as the excitation signal, the actual output response of the power amplifier is expanded by the Taylor series as follows:(2)$$\begin{array}{ll} s(t) & =v\cos wt*h_{Amp}(t)\\ & =(a_{0}+0.5a_{2}v^{2})+(a_{1}v+0.75a_{3}v^{3})\cos wt\\ & +0.5a_{2}v^{2}\cos2wt+0.25a_{3}v^{3}\cos3wt+\cdots \\ & =\sum_{i=0}^{n}a_{i}\cos iwt \end{array}$$

In Formula (2), vcoswt is the single-tone excitation signal, $h_{Amp}(t)$ is the actual system function of the power amplifier module, and $a_{i}$ is the amplitude of the i-th times frequency of the excitation signal by the power amplifier module. From the formula, it can be seen that the energy of the original signal is dispersed into harmonics, causing spectral growth in adjacent channels. The spectral growth caused by the power amplifier module leads to adjacent channel interference and also reveals its identity to the external environment.

Combining Formulas (1) and (2), the final output signal can be derived as follows:(3)$$S(t)=x(t)*h_{Amp}(t)\times e^{j(w+w^{\prime }(t))t+\varphi_{0}+\varphi (t)}$$

In Formula (3), the different features in the output signal S(t) do not interfere with each other and together form the emitter characteristics embedded in the transmitted signal. Different emitters have different carrier frequency offsets $f^{\prime }(t)$, phase noise and IQ offsets φ(t), and power amplifier nonlinear functions $h_{Amp}(t)$. However $w^{\prime }(t)$, φ(t), and $h_{Amp}(t)$ change over time and with external conditions, and these changes have a certain randomness. Therefore, finding a suitable mapping from the emitter sample space to the category space is very difficult, placing high demands on the performance of deep learning models.

### 2.2. Deep Learning with Limited Samples

In uncooperative scenarios, obtaining samples of transmitter signals is challenging, often only allowing access to a small portion of labeled data as training samples. Therefore, implementing specific emitter identification tasks with limited samples has a higher application value.

The optimization process of deep learning models is about learning the mapping $h:X_{s}\to Y_{s}$ from sample space $X_{s}$ to label space $Y_{s}$. However, for deep learning with limited samples, the training dataset only occupies a small part of the entire sample space, resulting in a difference between the distribution of the training set and the sample space. The training set can be represented as follows:(4)$$D^{train}=\{{\left(x_{i},y_{i}\right)}_{i}^{I},x_{i}\in X_{s},y_{i}\in Y_{s}\}$$

In Formula (4), I is the training sample set, and $x_{i}$ and $y_{i}$ represent the i-th sample and label. The deep learning model optimizes the model by minimizing the expected loss of the training dataset $D^{train}$:(5)$$R_{I}(h)=\underset{h\in H}{\min}\frac{1}{I}\sum_{x_{i}\in X_{s},y_{i}\in Y_{s}}l(h(x_{i}),y_{i})$$

In Formula (5), l is the loss function of the model, and H is the solution space. From Formula (4), it can be seen that the optimal model depends on the training dataset I and the solution space H. When the training dataset I is too small, the hypothesis space h available for the algorithm to choose is too large, leading to overfitting and poor generalization ability of the model.

The generalization problem with limited samples can be tackled from data augmentation and model perspectives.Increasing available sample quantity through data augmentation: Deep learning models map samples to a high-dimensional feature space $Z_{s}$. Since the dataset $D^{train}$ to a feature space $z_{train}$, there exists $z_{train}\subset Z_{s}$. Since augmented dataset $D^{aug}$ corresponds to a feature space $z_{aug}$, there exists as follows:(6)$$z_{train}\subset z_{train}\cup z_{aug}\subset Z_{s}$$Through data augmentation, constructing more data allows the model to learn closer to the real sample space, improving the network’s generalization ability.Reducing the solution space through contrastive learning: The goal of contrastive learning is to learn the mapping h through positive and negative samples, making the feature distance of the same category samples as close as possible and the feature distance of different category samples as far apart as possible:(7)$$l(f(x),f(x^{+}))>l(f(x),f(x^{-}))$$In Formula (7), the pair $x^{+}$ and x is a positive example, while the pair $x^{-}$ and x is a negative example. Through contrastive learning, the model tends to prefer algorithms that can minimize the contrastive loss and achieve better classification effects, improving the model’s generalization ability under limited samples.

## 3. The Proposed Method

### 3.1. The Framework of TWC-CNN

In order to achieve specific emitter identification tasks with limited samples, this paper proposes a TWC-CNN network framework that combines data augmentation and contrastive learning [44], as shown in Figure 2. The network can be divided into two parts: pre-training and fine-tuning. For the pre-training section, CWT was used as a data augmentation method to increase the sample size. The original samples X is augmented by adding additive white Gaussian noise (AWGN) with a signal-to-noise ratio of 10 to 30 and applying CWT to obtain samples $X_{1}$ and $X_{2}$, which are considered to be data of the same category. The input sample sizes for the CVNN and DCNN are 2×6000 and 256×6000. The size and time–frequency resolution of the CWT change with the transformation scale; in Section 4, this paper successively attempts identification results for CWT transformation scales ranging from 8 to 256. Unlike traditional enhanced data, $X_{1}$ and $X_{2}$ belonging to different modalities, using a weight-sharing encoder, make it difficult to simultaneously extract features from different modalities. Therefore, this paper does not use the traditional framework of contrastive learning with shared encoder weights. Instead, it extracts 1024-dimensional features $x_{i}$ and $x_{j}$ from $X_{1}$ and $X_{2}$ using CVNN and DCNN, respectively. $x_{i}$ and $x_{j}$ obtain 1024-dimensional features $h_{i}$, $h_{j}$, and $h_{k}$ through the fully connected network and vector-mask module and then obtain 128-dimensional features $z_{i}$ and $z_{j}$ through the projection head. Among them, $h_{k}$ and $h_{i}$ are used to calculate the cross-entropy loss, while $z_{i}$ and $z_{j}$ calculate the NT-Xent loss. The labels form a matrix as shown in Figure 2. The NT-Xent loss only treats $X_{1}$ and $X_{2}$ as positive samples, whereas other samples are considered as negative samples. At the same time, we calculate the cosine loss of $h_{i}$ and $h_{k}$ to correct the distribution of features of the sample $X_{1}$ and $X_{2}$. This article jointly trains a TWC-CNN network using cross-entropy loss, NT-Xent loss, and cosine loss. The NT-Xent loss reduces the intra-class distance between the enhanced data and the original data, while the cross-entropy loss supervises the classification loss between the enhanced data and the original data. At the same time, cosine loss is used to correct the feature distribution of the two modal data. It is found that controlling the ratio of NT-Xent loss and cross-entropy loss through hyperparameters λ can significantly enhance the accuracy of small sample transmitter recognition.

For the fine-tuning part, the parameters of the fully trained CVNN and DCNN are fixed, and then the extracted 1024-dimensional feature maps are concatenated and input into the classification network, which is a fully connected network. Fine-tune the classification network using the training dataset until the cross-entropy loss of the fully connected network converges.

#### Complex-Valued Neural Networks

Since signals are considered as complex numbers in the SEI task, with IQ representing the real and imaginary parts of the signal, respectively, we use a CVNN with complex-valued weights and biases to extract features from the IQ data while preserving the correlation between IQ signals.

The structure of the complex convolution block is shown in Figure 3b, where each block consists of complex convolution, activation, normalization, and Maxpooling. The complex convolution part can be expressed as follows:(8)$$\begin{array}{ll} cconv & =(r_{I}+jr_{Q})\times (conv1+j\times conv2)\\ & =conv1(r_{I})-conv2(r_{Q})+j(conv1(r_{Q})+conv2(r_{I}) \end{array}$$

In Equation (8), conv1 and conv2 are two convolution layers that do not share weights. The complex convolution block separates the IQ components of the complex signal before convolution and then recombines them afterward, achieving complex convolution. This approach maintains the connection between the real and imaginary parts of the IQ signal while relatively independently extracting the characteristics of the transmitter from the signal.

The CVNN used in this paper is shown in Figure 3a, consisting of 9 layers of complex convolution blocks, a flatten layer, and two fully connected layers. The structure of DCNN is consistent with that of CVNN, also comprising 9 layers of convolution blocks, a flatten layer, and two fully connected layers, except that the complex convolution blocks are replaced with regular convolution blocks.

### 3.2. Proposed Data Augmentation Method

#### 3.2.1. Traditional Data Augmentation Methods

To demonstrate the differences between our proposed method and traditional methods, we introduce four traditional data augmentation techniques and compare them with the CWT data enhancement method proposed in this paper. The augmented data waveforms using addnoise, rotation, flipping, and shifting methods are shown in Figure 4.

(1) Addnoise: Adding Gaussian white noise to the original signal is an effective way to enhance data. However, adding noise itself does not increase the information content of the original data and may reduce the performance of transmitter identification due to the presence of noise.(9)$$r^{\prime }=\left[\begin{array}{l} r_{I}\\ r_{Q} \end{array}\right]+N_{s}$$

In Formula (9), *N* represents Gaussian white noise. In this paper, we add random Gaussian white noise with a signal-to-noise ratio of 10–30 to the original data.

(2) Rotation: Keeping the amplitude unchanged, adding a random rotation angle to the original signal changes the distribution of IQ values.(10)$$r^{\prime }=\left[\begin{array}{l} \cos\varphi \\ -\sin\varphi \end{array}\right.\left.\begin{array}{l} \sin\varphi \\ \cos\varphi \end{array}\right]\left[\begin{array}{l} x_{I}\\ x_{Q} \end{array}\right]$$

In Formula (10), θ represents the rotation angle.

(3) Flipping: Data flipping is essentially a change in the intensity of the original signal, randomly converting it to positive or negative values to obtain flipped data.(11)$$r^{\prime }=\left[\begin{array}{ll} \pm 1 & 0\\ 0 & \pm 1 \end{array}\right]\left[\begin{array}{l} r_{I}\\ r_{Q} \end{array}\right]$$

(4) Shifting: Randomly shifting the signal changes the absolute position of each data point while maintaining their relative positions. Signal shifting disrupts causality but does not alter the characteristics of the transmitter.(12)$$r^{\prime }=\left[\begin{array}{l} i_{j},i_{j+1},\cdots \cdots ,i_{L},i_{1},i_{2},\cdots \cdots ,i_{j-1}\\ q_{j},q_{j+1},\cdots \cdots ,q_{L},q_{1},q_{2},\cdots \cdots ,q_{j-1} \end{array}\right]$$

In Formula (12), L represents the length of the original data, and d represents the length of the shift.

#### 3.2.2. Proposed CWT and FFT Data Augmentation Methods

For limited sample SEI tasks, increasing the number of training samples through data augmentation can significantly improve network performance. However, different data augmentation methods yield varying performance improvements. For example, common methods for IQ data augmentation such as addnoise, rotation, flipping, and shifting only introduce certain disturbances to the original signal without providing additional transmitter feature information. Excessive interference may even hinder normal network training.

Intuitively, we believe that the greater the difference between the augmented data and the original data in terms of feature vectors, the closer the representational feature space will be to the actual feature space, as shown in Figure 5. Therefore, we propose two data augmentation methods: FFT (fast Fourier transform) and CWT (continuous wavelet transform), which perform time–frequency transformations on signals to obtain samples that differ significantly from the original data.

(1) TFC: It is generally believed that transmitter information mainly resides in local oscillator offsets and phase noise. Applying FFT to IQ signals to obtain frequency domain information highlights the transmitter features of the signal. Performing FFT on the original IQ signals does not destroy transmitter features while providing new information, making it entirely feasible to use FFT as a method for data augmentation. This can be expressed as follows:(13)$$r^{\prime }=FFT(x)=\int_{-\infty }^{\infty }(r_{I}+jr_{Q})e^{-jwt}dt$$

In Formula (13), x is the original sample data, $r_{I}$ and $r_{Q}$ are in-phase and quadrature components of the original sample, respectively, and $x^{\prime }$ is the augmented data.

(2) TWC: Applying continuous wavelet transform to IQ signals allows for analysis at different time resolutions and frequency resolutions, yielding more detailed information.(14)$$r^{\prime }=CWT(x)=\frac{1}{\sqrt{\alpha }}\int_{-\infty }^{\infty }(r_{I}+jr_{Q})\psi (\frac{t-\tau }{\alpha })dt$$

In Formula (14), α is the scaling factor of the wavelet basis, which changes by controlling the value of α; τ is the time shift scale of the wavelet basis; ψ is the wavelet basis function. This paper selects the cmor complex wavelet as the wavelet basis.

The reason this paper chooses what appears to be a complex method of data augmentation is that we aim to obtain augmented data that differ significantly from the original samples. The intuition is that the greater the difference between input samples, the closer the model’s learning will be to the actual sample space, thereby enhancing the network’s generalization ability.

IQ and wavelet spectra after adding noise are shown in Figure 6. It is evident that our proposed methods differ significantly from traditional methods. The TFC and TWC data augmentation methods proposed in this paper not only change the original data’s feature probability distribution but also alter the data’s dimensionality, transitioning from vector-to-vector comparison to vector-to-image comparison. By converting one-dimensional IQ signals into two-dimensional wavelet spectrograms and using contrastive loss to calculate their similarity and train the network, it essentially becomes a type of multimodal network rather than just data augmentation.

### 3.3. Combining Contrastive Loss and Supervised Loss for Pre-Training Networks

Contrastive loss effectively addresses limited sample classification problems. In limited sample learning scenarios, where each class has very few labeled samples, traditional supervised learning methods often struggle to achieve good results. Contrastive learning constructs a self-supervised learning framework that learns discriminative feature representations without requiring large amounts of labeled data, thus alleviating the challenges of small-sample learning.

Since fine-tuning is a powerful technique but highly dependent on the quality and adaptability of the data, and small-sample learning lacks high-quality data, we realized that labeling training networks only during fine-tuning is insufficient. Thus, during the training phase, combining supervised contrastive loss [44] $L^{sup}$ is more suitable for limited sample contrastive learning:(15)$$L^{sup}=\sum_{i=1}^{2N}\frac{-1}{2N_{{\tilde{y}}_{i}}-1}\sum_{j=1}^{2N}I_{i\neq j}\cdot I_{{\tilde{y}}_{i}={\tilde{y}}_{j}}\cdot \log\frac{e^{\left(z_{i}\cdot z_{j}/\tau \right)}}{\sum_{k=1}^{2N}I_{i\neq k}\cdot e^{\left(z_{i}\cdot z_{k}/\tau \right)}}$$

In Formula (15), N is the number of images with the same label in a batch of input data; when i=j, ℝi≠k returns 1, otherwise returns 0.

We observed that the supervised contrastive loss can be divided into three parts: the contrastive loss between the original and augmented samples, the supervised loss of the original samples, and the supervised loss of the augmented samples, expressed as follows:(16)$$\begin{array}{c} L^{sup}=\sum_{i=1}^{N}\frac{-1}{2N_{{\tilde{y}}_{i}}-1}(2\log\frac{e^{\left(z_{i}\cdot z_{j}/\tau \right)}}{\sum_{k=1}^{2N}I_{i\neq k}\cdot e^{\left(z_{i}\cdot z_{k}/\tau \right)}}{|}_{j=i+N}\\ +\sum_{j=1}^{N}I_{i\neq j}\cdot I_{{\tilde{y}}_{i}={\tilde{y}}_{j}}\cdot \log\frac{e^{\left(z_{i}\cdot z_{j}/\tau \right)}}{\sum_{k=1}^{2N}I_{i\neq k}\cdot e^{\left(z_{i}\cdot z_{k}/\tau \right)}}\\ +2\sum_{j=1}^{N}I_{i\neq j}\cdot I_{{\tilde{y}}_{i}={\tilde{y}}_{j}}\cdot \log\frac{e^{\left(z_{i}\cdot z_{j}/\tau \right)}}{\sum_{k=1}^{2N}I_{i\neq k}\cdot e^{\left(z_{i}\cdot z_{k}/\tau \right)}}) \end{array}$$

In supervised contrastive loss, the proportions of these three parts are fixed. However, intuitively, we should prioritize clustering original samples and augmented samples together first and then use the contrastive loss to bring the distance between positive original samples and positive augmented samples closer. Specifically, changing the proportion of contrastive loss between original and augmented samples can significantly improve the classification effect in small-sample learning, as reflected in Section 4 experiments showing how the contrastive loss ratio coefficient λ affects classification results.

To separate $L^{sup}$ at the network level, we rewrite it as $L^{TWC}$:(17)$$\begin{array}{c} L^{TWC}=\lambda \cdot (\sum_{i=1}^{N}-\log\frac{e^{\left(sim(z_{i}\cdot z_{j})/\tau \right)}}{\sum_{k=1}^{2N}I_{i\neq k}\cdot e^{\left(sim(z_{i}\cdot z_{k})/\tau \right)}})\\ +\sum_{i=1}^{N}-\frac{1}{N}\sum_{j=1}^{N}y_{i}\log\frac{e^{z_{1}^{i}}}{\sum_{k=1}^{N}e^{z_{1}^{k}}}\\ +\sum_{i=1}^{N}-\frac{1}{N}\sum_{j=N}^{2N}y_{i}\log\frac{e^{z_{2}^{i}}}{\sum_{k=1}^{N}e^{z_{2}^{k}}}\\ =\lambda \cdot L^{NT-Xent}+L^{CE-O}+L^{CE-Aug} \end{array}$$

In Formula (17), where $y_{i}$ is the corresponding label and c is the number of classes. Since the two encoders in TWC-CNN proposed in this paper do not share weights, it is necessary to train both encoders using supervised loss and contrastive loss.

In NT-Xent loss, the numerator part maximizes the similarity between positive samples, while the denominator minimizes the similarity between negative samples. The formula for NT-Xent loss [38] is as follows:(18)$$L^{NT-Xent}=\lambda \cdot (\sum_{i=1}^{N}-\log\frac{e^{\left(sim(z_{i}\cdot z_{j})/\tau \right)}}{\sum_{k=1}^{2N}I_{i\neq k}\cdot e^{\left(sim(z_{i}\cdot z_{k})/\tau \right)}})$$

This paper calculates NT-Xent loss for $z_{i}$ and $z_{i}$ to measure the feature differences between original and augmented data. The cross-entropy loss is as follows:(19)$$L^{CE-O}=\sum_{i=1}^{N}-\frac{1}{N}\sum_{j=1}^{N}y_{i}\log\frac{e^{z_{1}^{i}}}{\sum_{k=1}^{N}e^{z_{1}^{k}}}$$(20)$$L^{CE-Aug}=\sum_{i=1}^{N}-\frac{1}{N}\sum_{j=N}^{2N}y_{i}\log\frac{e^{z_{2}^{i}}}{\sum_{k=1}^{N}e^{z_{2}^{k}}}$$

This paper calculates cross-entropy loss for $z_{1}$ and $z_{2}$, representing the classification losses for original and augmented data, respectively.

### 3.4. Correcting the Feature Probability Distribution After Data Augmentation Through Vector-Mask Module

After training the network with contrastive and cross-entropy losses, the feature vectors for the same category have consistent probability distributions. However, for original and augmented samples, the consistency of their feature vector probability distributions is not enough since augmented samples originate from original samples; hence, their feature vector probability distributions should be as similar as possible.

We project $h_{j}$ onto $h_{k}$ using a masked projection head. The similarity between feature vectors $h_{i}$ and $h_{k}$ extracted by CVNN and DCNN is measured by cosine distance, represented as follows:(21)$$L_{cosine}=\left\{\begin{array}{l} 1-\frac{h_{i}\cdot h_{k}}{\left|h_{i}\right|\times \left|h_{k}\right|},\;ify=1\\ \max(0,\frac{h_{i}\cdot h_{k}}{\left|h_{i}\right|\times \left|h_{k}\right|}-\mathrm{margin}),ify=-1 \end{array}\right.$$

In Formula (21), $h_{i}$ and $h_{k}$ are features extracted by CVNN, and margin represents the loss threshold, exceeding the loss threshold, $L_{cosine}$ which becomes 0. After incorporating the vector-mask module, the $L^{TWC}$ becomes(22)$$L^{TWC}=\lambda \cdot L^{NT-Xent}+L^{MSE-O}+L^{MSE-Aug}+L^{cosine}$$

This paper pre-trains the TWC-Net network to achieve transmitter identification in small-sample scenarios.

## 4. Experimental Setup and Results

### 4.1. Experimental Setup

We evaluated the TWC-CNN method proposed in this paper using open-source WiFi datasets and open-source ADS-B datasets from refs. [45,46]. The ADS-B (Automatic Dependent Surveillance–Broadcast) dataset originates from civil and military aircraft’s ADS systems, transmitting signals containing information such as the aircraft’s position, altitude, speed, heading, and identification number, propagated through wireless channels. The WiFi dataset comes from a communication system built with USRP X310 transmitters (Ettus Research, Austin, TX, USA) and USRP B210 receivers (National Instruments, Austin, TX, USA), where the transmitted signals are random data generated by MATLAB (R2021b) WLAN System toolbox compliant with the IEEE 802.11a standard [47], propagated through wireless channels. This paper selected ADS-B data from 10 transmitters and WiFi datasets from 16 transmitters to test the classification performance of the TWC-CNN method. Due to different IQ sampling rates for the two datasets, to obtain sufficient specific emitter information, each ADS-B dataset entry is 4800 samples long, with a total of 3080 entries in the training set and 1000 entries in the test set; each WiFi dataset entry is 6000 samples long, with a total of 3080 entries in the training set and 16,004 entries in the test set. It should be noted that to simulate actual specific emitter identification tasks, the data volume for each category varies. This paper constructed five datasets with varying sample sizes to evaluate the specific emitter identification performance of the proposed TWC-CNN network under small sample conditions. These datasets contain {1%, 5%, 10%, 15%, 20%} of the total data volume, respectively, ensuring at least one sample per category. We used the PyTorch (version 2.5.1) framework to build our network. The training of the TWC-CNN network consists of pre-training and fine-tuning phases. In the pre-training phase, the LARS optimizer is used to optimize the network parameters with a learning rate of 0.2, a batch size of 64, and the hyperparameter λ controlling the contrastive loss set to 0.001, performing 300 iterations. In the fine-tuning phase, the Adam optimizer is used with a learning rate of 0.001, a batch size of 256, and 120 iterations. Our experiments utilized four NVIDIA GeForce RTX 2080Ti GPUs (NVIDIA Corporation, Santa Clara, CA, USA).

### 4.2. Introduction to Comparative Methods

This paper compares the proposed TFC-CNN and TWC-CNN with six state-of-the-art methods: CVNN [48], DRCN [49], SSRCNN [50], TripleGAN [51], MAT-PA [37], and SimMIM [52]. For fairness, all six methods and the proposed TWC-CNN use the same CVNN as the backbone network, and they are trained and tested with the same data, training iterations, and learning rates. Additionally, since SimMIM, SSRCNN, and TripleGAN can use unlabeled data to enhance network performance, this paper focuses on small-sample specific emitter identification tasks, no additional unlabeled data are provided.

### 4.3. Comparison with Other Methods

The number of samples in the training dataset directly affects the identification performance of SEI. However, in practical applications, it is difficult to obtain a large amount of target data. Therefore, the proposed SEI method can train and identify target specific emitters with a small amount of data, indicating that the proposed SEI method can be applied in real-world scenarios. The SEI identification based on TFC-CNN and TWC-CNN and six comparison methods on the WiFi dataset and ADS-B dataset are shown in Table 1. In both the ADS-B dataset and WiFi dataset, our proposed TWC-CNN method achieves the best identification performance under all sample sizes. For the ADS-B dataset, when the sample size is only 1% of the total training samples, the accuracy rates of the other six methods range from 10% to 35%, whereas the identification accuracy rates of the TFC-CNN and TWC-CNN methods proposed in this paper exceed 35%, reaching 42.90%. For the WiFi dataset, the SEI method based on TWC-CNN exhibits significant performance advantages. When the sample size accounts for 1% of the total training samples, the identification accuracy rates of the other methods are below 20%, while the SEI method based on TWC-CNN proposed in this paper achieves a recognition effect of 52.00%, significantly outperforming the six comparison methods.

Additionally, when the sample size of the WiFi dataset is 5%, the recognition rates of the other six methods are below 40%, but the identification accuracy of our proposed method reaches 96.99%.

### 4.4. Visualization of Feature Data

To more intuitively demonstrate the feature extraction performance of the proposed TWC-CNN and TFC-CNN on specific emitter characteristics, we reduced the feature vectors extracted by TWC-CNN, TFC-CNN, CVNN, DRCN, SimMIM, SSRCNN, TripleGAN, and MAT-PA to two dimensions using T-SNE with 10% of the WiFi dataset. The two-dimensional data were used as the horizontal and vertical coordinates to display the feature vectors, as shown in Figure 7. As can be seen from Figure 7, the T-SNE visualization of TWC-CNN and TFC-CNN is better than that of the six comparison networks. In the T-SNE plots of SEI based on TWC-CNN and TFC-CNN, the features of different categories are completely separated, while the data belonging to the same category aggregate together, with only partial categories having slight overlaps. In contrast, the T-SNE plots based on CVNN, DRCN, SSRCNN, TripleGAN, and MAT-PA methods show overlapping categories that are not fully separated, making it difficult to distinguish the data of the 16 categories. Among them, since SimMIM did not use additional unlabeled data for pre-training and its method of jointly fine-tuning the encoder and classifier parameters can easily lead to overfitting, the T-SNE plot of SimMIM shows all categories overlapping, making it difficult to distinguish.

### 4.5. Comparison of Training Time

In practical applications of specific emitter identification, the computation time of the network represents the target detection time. However, since this paper focuses on recognition performance under small samples, even with larger networks, the iteration time for a single training sample is shorter due to the limited number of training samples. The training times of the proposed SEI based on TWC-CNN and TFC-CNN compared to six benchmark methods are shown in Table 2. Due to the need to calculate the wavelet transform of IQ signals and input the wavelet spectrum data into the network, the computation time of the TWC-CNN method significantly increases. For instance, with 1% of the WiFi dataset, the single data iteration time of the TWC-CNN method is 2.14 s. TFC-CNN only adds one additional FFT and noise calculation in data augmentation. However, since TFC-CNN uses two separate networks to extract features from IQ signals and spectrum data, and additionally calculates vector-mask-based similarity and contrastive loss, its operation time is longer than that of CVNN. Among the six comparison methods, TripleGAN and DRCN increase training time due to the addition of encoder, decoder, discriminator, and other network structures, making it longer than TFC-CNN. TWC-CNN has similar training times to SSRCNN and MAT-PA due to the use of data augmentation. Additionally, since SimMIM’s unlabeled data pre-training was canceled, the training time does not extend due to increased data volume.

### 4.6. Ablation Experiment

To analyze the performance improvement effect of each part of the SEI method based on TWC-CNN, we conducted ablation experiments on the TWC-CNN network. The results are shown in Table 3, and w/o TWC represents the elimination of time–wavelet spectrum data augmentation for TWC-CNN, replaced by adding random noise as network input; w/o CELoss represents the elimination of the supervised training process during the pre-training stage of TWC-CNN, making it a typical contrastive learning network; w/o vector-mask represents the elimination of the contrastive learning part, making TWC-CNN a typical supervised network; w/o NT-Xent loss represents the elimination of feature vector similarity supervision between time and wavelet spectra. In ablation experiments, TWC-CNN achieved the highest accuracy across all datasets and data ratios. Only when all network components work together higher recognition accuracy can be obtained. Eliminating any part of the TWC-CNN network leads to performance degradation. It should be noted that w/o NT-Xent loss is prone to loss explosion during the pre-training stage, as without contrastive loss to separate feature vectors of different categories, all feature vectors tend to converge, causing the loss to approach infinity.

### 4.7. The Impact of Hyperparameters on the Performance of the TWC-CNN Network

Different network hyperparameters directly affect the performance of the network. The main hyperparameters for the SEI method based on TWC-CNN proposed in this paper are the coefficient λ of the contrastive loss and the scale of CWT, whose impact on network performance is shown in Figure 8 and Figure 9.

Figure 8 displays the recognition accuracy of TWC-CNN and TFC-CNN with coefficient λ set at 0.0001, 0.001, 0.01, 0.1, and 0.5 under different sample sizes of the WiFi dataset and ADS-B dataset. Clearly, for all values of coefficient λ, as the proportion of samples increases, the network’s recognition accuracy gradually improves. Specifically, when λ = 0.001, both TWC-CNN and TFC-CNN achieve optimal recognition performance across almost all parameter quantities and datasets. During network training, the optimal choice of the contrastive loss coefficient is closely related to the structure of the target dataset; compared to the ADS-B dataset, the WiFi dataset exhibits more robustness in selecting the coefficient λ.

Figure 9 shows the recognition accuracy of TWC-CNN under different CWT scales on the ADS-B dataset, illustrating that as the proportion of training samples to all samples increases, the recognition performance significantly improves. However, an increase in the CWT scale does not necessarily enhance the performance of TWC-CNN; sometimes it even decreases. This is because while an increase in the CWT scale enriches the emitter features of the input data, making it easier for the network to extract key information, it also significantly increases the size of the input data. A larger feature distribution space makes it difficult for the network to focus on the local features of all data. As can be seen from Figure 9, when the scale is 128, TWC-CNN achieves higher recognition performance. Additionally, in Figure 9, FFT is considered a single transform of CWT and included in the data for comparison.

### 4.8. The Impact of Different Data Augmentation Effects on Experimental Results

In scenarios with limited samples, data augmentation can provide more sample data, reducing the overfitting caused by limited samples. This paper uses four classic data augmentation methods—random noise, random rotation, random flipping, and random shifting—to compare with the TFC and TWC data augmentation methods proposed in this paper, verifying the recognition performance of TFC and TWC under different data ratios on ADS-B and WiFi datasets. The results are shown in Table 4 and Figure 10. It is evident that the time-domain wavelet spectrum data augmentation method proposed in this paper has the best recognition accuracy for both WiFi and ADS-B datasets at different data ratios. This is because the TWC method not only augments the existing data but also obtains richer time–frequency information through time–frequency transformation. While traditional data augmentation methods can generate new samples for training the network, they may not bring new information and could potentially damage specific emitter features during the transformation process, leading to a decrease in recognition accuracy.

As seen in Figure 10, as the amount of data increases, the final recognition accuracy of all data augmentation methods will improve. Under 5% of the WiFi dataset, the recognition accuracy of other methods is below 85%, while the TFC and TWC methods proposed in this paper exceed 95%, demonstrating the significant advantages of the TFC and TWC methods. At 20% of the WiFi and ADS-B datasets, the recognition accuracy using random shifting approaches that of TFC and TWC. This is because when the data volume already meets the requirements for network training, the SEI accuracy of different data augmentation methods tends to stabilize.

## 5. Conclusions

In this paper, we propose a novel TWC-CNN-based SEI recognition method that achieves good recognition performance even with limited samples. Specifically, this method consists of data augmentation based on time-domain wavelet spectrum transformation and a feature extraction network based on the contrastive learning framework. First, the contrastive data in the time-domain wavelet spectrum is obtained through wavelet transformation, and then specific emitter features are extracted separately by the complex-valued neural networks and deep convolutional networks. Afterward, the TWC-CNN network and TFC-CNN are pre-trained together using cosine loss, contrastive loss, and cross-entropy. Finally, the network is fine-tuned and classified. This method was evaluated on open-source ADS-B and WiFi datasets and compared with six state-of-the-art specific emitter recognition networks. Experimental results show that our proposed TWC-CNN-based SEI method has better recognition accuracy with limited samples. When the sample size is 5%, the accuracy on the WiFi dataset reached 96.99%, and visualization experiments also verified that TWC-CNN’s resolution effect on specific emitter features is superior to the other six methods.

In addition, during the course of this research, we discovered that even when training networks on datasets with sufficient samples, the pre-existing label distribution within the dataset has a significant impact on the final classification results. For example, if samples from one category constitute a larger proportion of all samples while those from other categories are less represented, the network trained on such data tends to favor the more prevalent category in its identification outcomes, leading to a decrease in overall accuracy. Therefore, how to achieve high recognition accuracy for SEI tasks under conditions of sample imbalance will be the focus of our subsequent work.

## Figures and Tables

**Figure 1 sensors-25-00648-f001:**
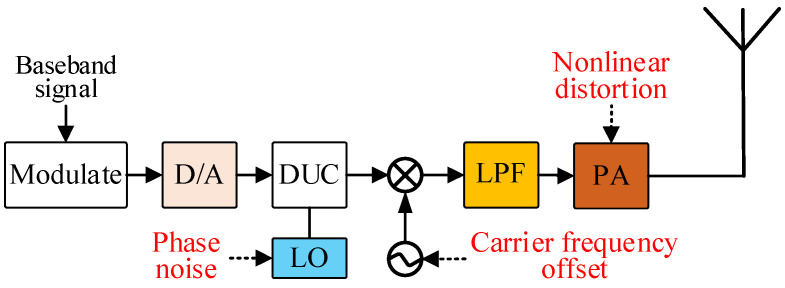
The formation principle of the radio frequency fingerprint in the emitter.

**Figure 2 sensors-25-00648-f002:**
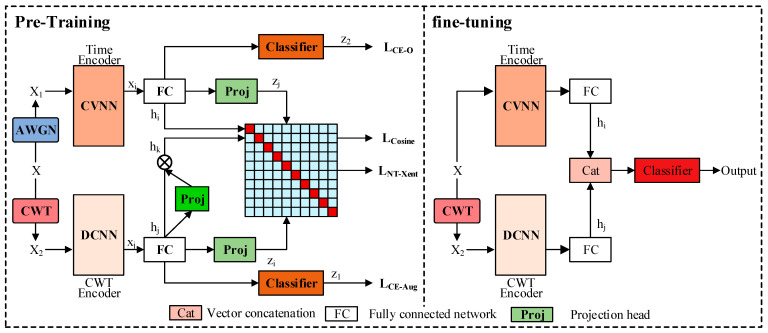
Structure of TWC-CNN.

**Figure 3 sensors-25-00648-f003:**
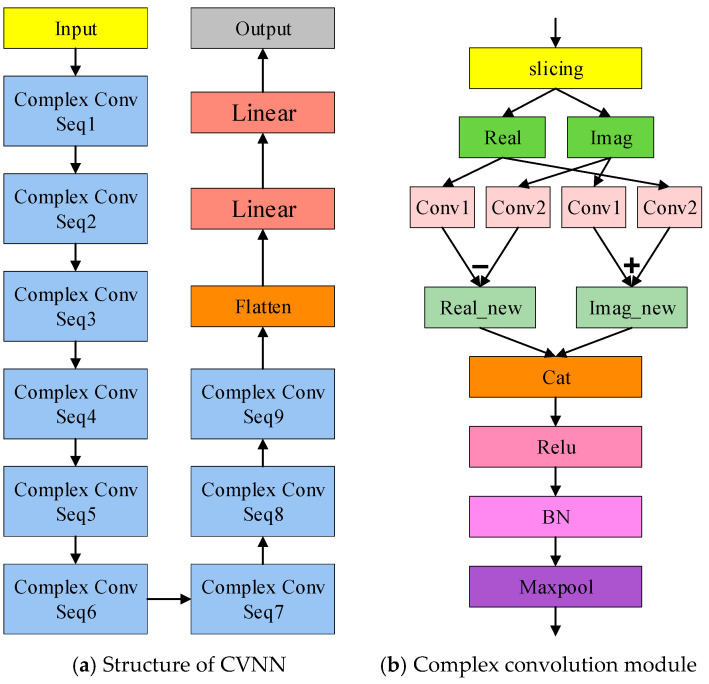
CVNN and complex convolutional module.

**Figure 4 sensors-25-00648-f004:**
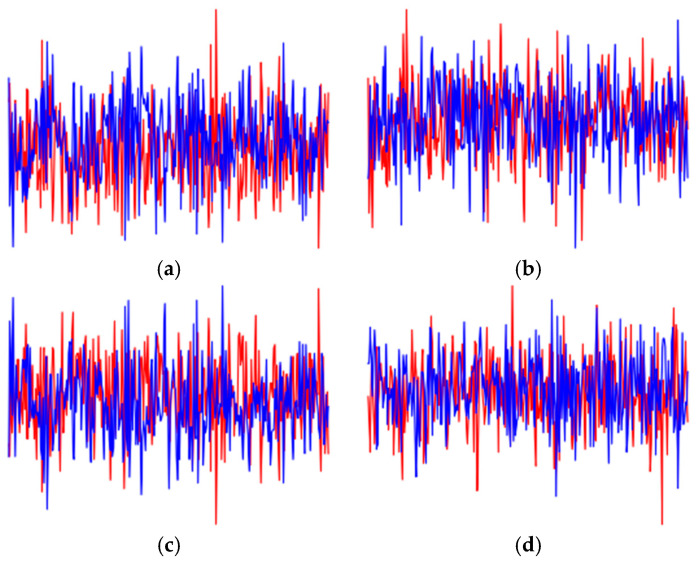
IQ waveform after data augmentation, The red curve represents the in-phase signal, and the blue curve represents the quadrature signal. (**a**) addnoise, (**b**) rotation, (**c**) flipping, and (**d**) shifting.

**Figure 5 sensors-25-00648-f005:**
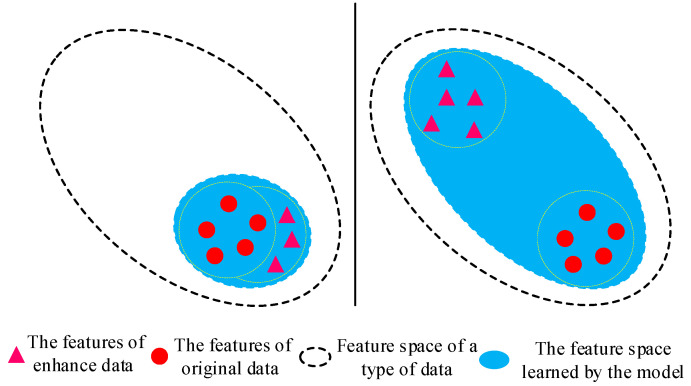
The larger the difference between the augmented data and the original data in terms of feature vectors, the closer the representational feature space is to the actual feature space.

**Figure 6 sensors-25-00648-f006:**
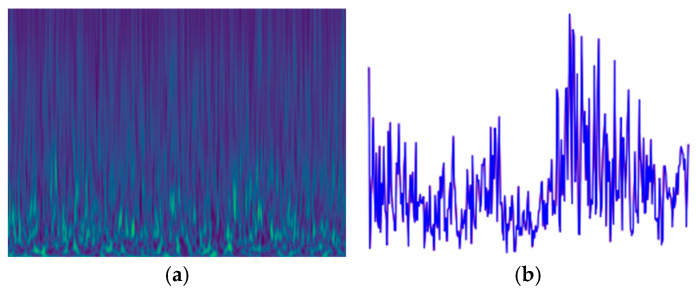
IQ and wavelet spectra after adding noise: (**a**) wavelet spectra and (**b**) IQ.

**Figure 7 sensors-25-00648-f007:**
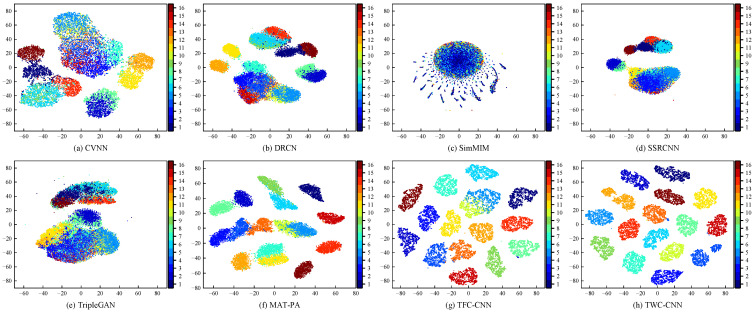
Visualization of specific emitter features extracted by the proposed TWC-CNN and TFC-CNN methods, compared with CVNN, DRCN, SimMIM, SSRCNN, TripleGAN, and MAT-PA methods, on the WiFi dataset when the sample size accounts for 10% of the total training samples.

**Figure 8 sensors-25-00648-f008:**
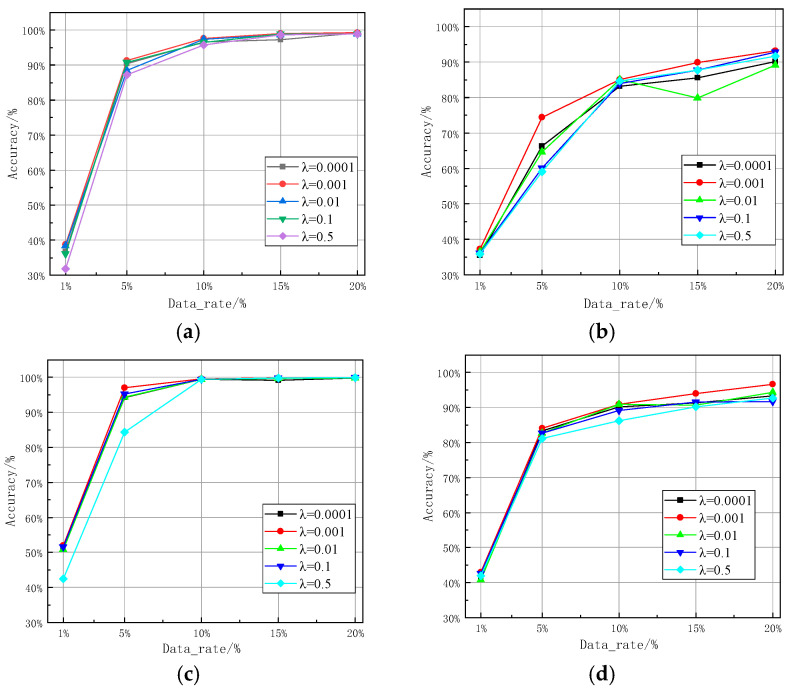
The impact of the contrastive loss coefficient λ on recognition accuracy: (**a**) TFC-CNN in the WiFi dataset; (**b**) TFC-CNN in the ADS-B dataset; (**c**) TWC-CNN in the WiFi dataset; and (**d**) TWC-CNN in the ADS-B dataset.

**Figure 9 sensors-25-00648-f009:**
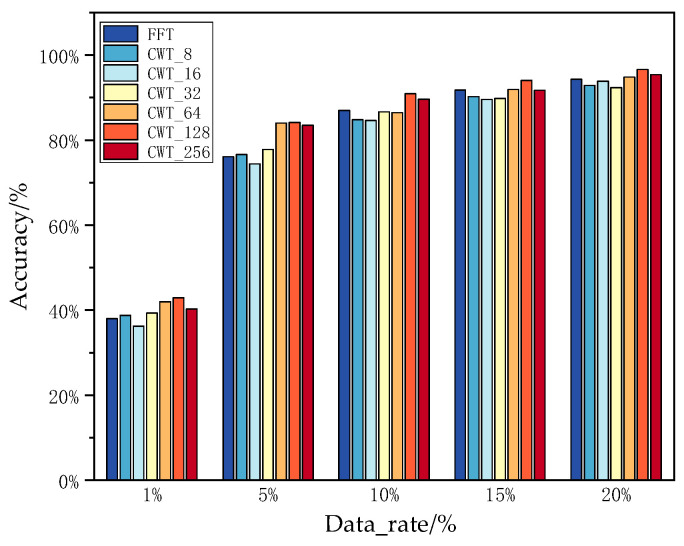
The impact of the scale of CWT on recognition accuracy.

**Figure 10 sensors-25-00648-f010:**
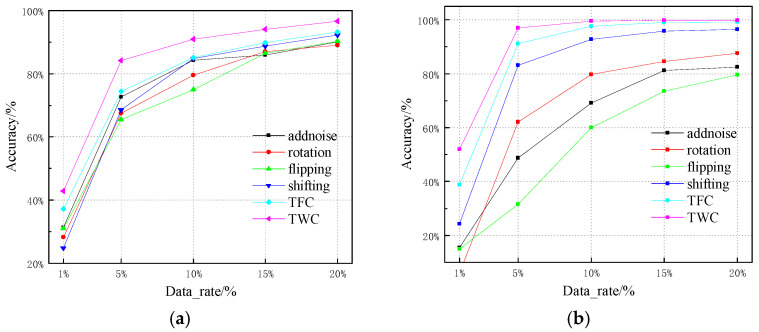
EI recognition accuracy of different data augmentation methods: (**a**) ADS-B dataset and (**b**) WiFi dataset.

**Table 1 sensors-25-00648-t001:** Identification accuracy of SEI based on TWC-CNN for the ADS-B dataset and WiFi dataset.

Methods	ADS-B					WiFi				
	1%	5%	10%	15%	20%	1%	5%	10%	15%	20%
CVNN [48]	28.50%	67.20%	82.30%	92.50%	95.60%	13.98%	31.95%	51.94%	75.35%	95.67%
DRCN [49]	26.50%	70.90%	88.80%	90.80%	95.80%	13.50%	29.42%	56.11%	81.28%	87.18%
SimMIM [52]	30.70%	81.30%	90.10%	92.50%	94.90%	19.40%	38.68%	45.46%	65.41%	77.30%
SSRCNN [50]	16.40%	40.40%	90.30%	92.90%	95.00%	6.22%	12.32%	60.85%	63.27%	81.51%
TripleGAN [51]	30.60%	50.00%	55.30%	67.20%	72.40%	13.78%	30.45%	64.81%	88.84%	94.96%
MAT-PA [37]	33.10%	71.90%	73.70%	92.90%	93.40%	12.43%	21.99%	59.50%	98.17%	99.80%
TFC-CNN	37.20%	74.40%	85.10%	89.90%	93.20%	38.80%	91.25%	97.58%	98.99%	99.23%
TWC-CNN	42.90%	84.10%	90.90%	94.00%	96.60%	52.00%	96.99%	99.49%	99.82%	99.89%

**Table 2 sensors-25-00648-t002:** Comparison of training times for different networks.

Ratio	CVNN	DRCN	SimMIM	SSRCNN	TripleGAN	MAT-PA	TFC-CNN	TWC-CNN
1%	0.15 s	1.53 s	0.14 s	0.37 s	1.22 s	0.63 s	0.44 s	2.14 s
5%	0.74 s	7.74 s	0.73 s	2.05 s	3.21 s	3.15 s	1.26 s	5.15 s
10%	1.48 s	15.36 s	1.37 s	3.93 s	6.51 s	6.24 s	2.24 s	6.42 s
15%	2.16 s	23.16 s	2.11 s	5.81 s	9.80 s	9.03 s	3.17 s	7.71 s
20%	2.92 s	30.97 s	2.86 s	7.54 s	13.08 s	12.65 s	4.22 s	9.31 s

**Table 3 sensors-25-00648-t003:** Ablation experiments on the proposed TWC-CNN under the WiFi dataset.

Methods	CE Loss	TWC	NT-Xent Loss	Vector-Mask	ADS-B		WiFi	
					5%	20%	15%	20%
TFC-CNN w/o CELoss		✓	✓	✓	72.10%	90.40%	72.93%	97.40%
TFC-CNN w/o TWC	✓		✓	✓	72.70%	90.10%	48.70%	82.50%
TFC-CNN w/o NT-Xent loss	✓	✓		✓	10.00%	10.00%	10.00%	10.00%
TFC-CNN w/o vector-mask	✓	✓	✓		65.10%	89.70%	69.90%	98.04%
TWC-CNN	✓	✓	✓	✓	84.10%	96.60%	96.99%	99.89%

The checkmark (✓) indicates that this module was used in the ablation experiment.

**Table 4 sensors-25-00648-t004:** SEI recognition accuracy of different data augmentation methods.

Method	ADS-B		WiFi	
	5%	20%	5%	20%
Addnoise	72.70%	90.10%	48.70%	82.50%
Rotation	67.50%	89.00%	62.16%	87.60%
Flipping	65.50%	90.30%	31.60%	79.57%
Shifting	68.60%	92.30%	83.07%	96.51%
TFC	74.40%	93.20%	91.25%	99.23%
TWC	84.10%	96.60%	96.99%	99.89%

## Data Availability

Data are contained within the article.

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
