# Peer review of "Specific Emitter Identification Method for Limited Samples via Time–Wavelet Spectrum Consistency"

_sensors, 2025, doi:10.3390/s25030648_

Round 1

Reviewer 1 Report

Comments and Suggestions for Authors

This paper proposes a specific emitter identification method for limited samples via time-wavelet spectrum consistency. There are some problems in the paper, and the comments are listed below.

1.      The network of the paper mainly refers to the paper: P. Khosla, P. Teterwak, C. Wang, A. Sarna, Y. Tian, P. Isola, A. Maschinot, C. Liu, and D. Krishnan. Supervised Contrastive Learning. Advances in neural information processing systems 2020, 18661–18673. But this paper and the other papers that corresponding to this paper are not listed as references.

2.      In the introduction, the term CWT is not introduced. In line 105, it is said the CWT provides different transformed domain data, how does it work? 

3.      In figure 2, the terms AWGN, Proj, CWT, Lsupcon and Cat are all not introduced and explained. The inputs and outputs of CVNN and DCNN are not well explained in the paper.

4.      There are several written mistakes, for example, the format error in line 180, the DCNN is written as CNN in line 237, etc.

5.      In equation 5, the type of loss is confused, is it cosine loss or NT-Xent loss?

6.      In figure 6, several methods are compared, but the proposed TWC-CNN doesn’t exist. The coordinate labeling should be added in the figures.

7.      In the simulation, the loss LNT-Xent doesn’t introduced, the sizes of the data that input and output from CVNN and DCNN are not given. The simulation settings should be carefully checked.

8.      There are many format mistakes in the reference, which should be carefully checked.

Reviewer 2 Report

Comments and Suggestions for Authors

The manuscript proposes a method based on Time Wavelet Spectrum Consistency Neural Networks (TWC-CNN). The approach is compared against mainstream algorithms using publicly available datasets and experimental evaluations. The results demonstrate excellent classification performance, particularly in scenarios with small sample sizes. My recommendation for this article is to accept it with minor revisions. To further improve this manuscript, I have the following suggestions:

1. It is recommended that the author enrich the methods section by incorporating additional mathematical formulas to describe the proposed method in greater detail. This would improve the clarity and precision of the technical explanations and facilitate a deeper understanding for readers.

2. A discrepancy has been noted in the content of the figures, such as the term "TCC net," which does not appear elsewhere in the manuscript. The author is advised to provide an explanation or revise the figures to ensure consistency between the visuals and the main text.

Reviewer 3 Report

Comments and Suggestions for Authors

1- Please explain the research gap better in the abstract.

2- The scientific name of the term is included with the abbreviation when it first appears. Please continue to apply this command such as the abbreviation SEI.

3- There is no need to mention the percentage of achievement or the numerical values ​​of the results, as it is considered a contribution from the researcher, as indicated in point 3 in the contributions.

4- Add a paragraph to the end of the introduction that explains the structure of the research with all its sections.

5- The title of section 2 is the same as the title of section 2.2.

6- Discuss the results in tables and figures.

7- Explain future work in more detail.

8-Explain the proposed algorithm in brief points.

Round 2

Reviewer 3 Report

Comments and Suggestions for Authors

All comments done.